# Cloud Removal in the Tibetan Plateau Region Based on Self-Attention and Local-Attention Models

**DOI:** 10.3390/s24237848

**Published:** 2024-12-08

**Authors:** Guoqiang Zheng, Tianle Zhao, Yaohui Liu

**Affiliations:** School of Surveying and Geo-Informatics, Shandong Jianzhu University, Fengming Road, Jinan 250101, China; gqzheng@sdjzu.edu.cn (G.Z.); 2022165119@stu.sdjzu.edu.cn (T.Z.)

**Keywords:** deep learning, cloud removal, attention model, multi-scale fusion module, Sentinel-2

## Abstract

Optical remote sensing images have a wide range of applications but are often affected by cloud cover, which interferes with subsequent analysis. Therefore, cloud removal has become indispensable in remote sensing data processing. The Tibetan Plateau, as a sensitive region to climate change, plays a crucial role in the East Asian water cycle and regional climate due to its snow cover. However, the rich ice and snow resources, rapid snow condition changes, and active atmospheric convection in the plateau as well as its surrounding mountainous areas, make optical remote sensing prone to cloud interference. This is particularly significant when monitoring snow cover changes, where cloud removal becomes essential considering the complex terrain and unique snow characteristics of the Tibetan Plateau. This paper proposes a novel Multi-Scale Attention-based Cloud Removal Model (MATT). The model integrates global and local information by incorporating multi-scale attention mechanisms and local interaction modules, enhancing the contextual semantic relationships and improving the robustness of feature representation. To improve the segmentation accuracy of cloud- and snow-covered regions, a cloud mask is introduced in the local-attention module, combined with the local interaction module to modulate and reconstruct fine-grained details. This enables the simultaneous representation of both fine-grained and coarse-grained features at the same level. With the help of multi-scale fusion modules and selective attention modules, MATT demonstrates excellent performance on both the Sen2_MTC_New and XZ_Sen2_Dataset datasets. Particularly on the XZ_Sen2_Dataset, it achieves outstanding results: PSNR = 29.095, SSIM = 0.897, FID = 125.328, and LPIPS = 0.356. The model shows strong cloud removal capabilities in cloud- and snow-covered areas in mountainous regions while effectively preserving snow information, and providing significant support for snow cover change studies.

## 1. Introduction

In recent years, remote sensing imagery has garnered widespread attention due to its promising applications in fields such as earth observation and environmental monitoring [1]. However, on average, 66% of the earth’s sky is covered by clouds annually [2]. Clouds seriously affect the quality of optical remote sensing images, which not only obscures the surface information but also destroys the spectral and texture information of the ground surface, which greatly reduces the reliability of remote sensing images and increases the difficulty of data processing [3]. If imagery is discarded, it will result in wasted data and even no imagery data available in some cloudy areas. Therefore, in order to improve the availability of optical remote sensing image data, it is a necessary and effective preprocessing method to reconstruct the cloud coverage area. In high-altitude regions, however, the alternating coverage of clouds and snow poses additional challenges. Snow often exhibits similar reflectance characteristics and texture information to clouds, which significantly impairs the accuracy of cloud detection, making cloud removal in these areas more challenging than in other scenarios. To better understand the challenges of cloud removal on the Tibetan Plateau and the progress in research, it is essential to recognize the limitations of single-temporal cloud removal methods when dealing with images that have extensive cloud coverage. While these methods can generate relatively stable cloud-free images from individual cloud-covered images, the lack of sufficient ground information in cases of widespread cloud coverage makes it more difficult to produce cloud-free imagery. In such situations, the performance of single-temporal cloud removal methods is constrained, making them inadequate for practical applications. In recent years, researchers have been exploring the use of multi-temporal satellite images for cloud removal and have achieved significant results. Multi-temporal cloud removal methods utilize multiple cloud-covered images from the same location as input, integrating spatial and temporal information to generate cloud-free images. For example, Li et al. proposed a convolutional neural network (CNN)–based autoencoder for cloud removal using multi-temporal satellite images [4]. Jin et al. approached the cloud removal problem as a conditional image synthesis challenge and proposed a spatiotemporal generative network for cloud removal [5].

The Tibetan Plateau, known as one of the regions with the highest snowfall in the world, presents significant challenges for cloud removal due to its complex climatic features and geographical conditions. With an average elevation exceeding 4000 m, the Tibetan Plateau is the largest and highest plateau globally, and its sensitivity to climate change makes monitoring snow cover changes crucial [6]. As the “Water Tower of Asia”, the melting snow on the Tibetan Plateau provides water for major rivers such as the Yangtze, Yellow, Yarlung Tsangpo, Indus, and Ganges. However, traditional field surveys and meteorological observation methods are difficult to implement in this region, complicating the accurate measurement of snow changes. In this context, satellite remote sensing technology has gradually become the mainstream method for snow monitoring due to its advantages of continuity, broad coverage, high temporal resolution, and reliable data quality, effectively compensating for the limitations of ground-based meteorological station data. However, the annual average cloud cover over the Tibetan Plateau reaches a maximum of approximately 87% [7] and the high spectral similarity between clouds and snow further increases the classification errors in the images. These two factors significantly hinder the utilization and analysis of remote sensing images, particularly when clouds and snow are mixed, making cloud removal from the imagery especially challenging.

## 2. Related Works

### 2.1. Cloud and Snow Segmentation

Cloud detection in cloud- and snow-covered areas is a crucial step in the cloud removal process. By identifying the percentage of cloud cover, cloud detection can serve as an important indicator of image quality and data availability [8]. It helps extract useful data, enhances the storage and transmission efficiency of image data, and provides essential products during the preprocessing stage, maximizing the utility of remaining cloud-free areas and improving the applicability of images, especially in cloud- and snow-covered [9]. However, currently, only a few satellite products (such as Landsat, MODIS, and Sentinel-2) provide corresponding cloud mask products [10]. Therefore, developing fast and straightforward cloud detection algorithms is particularly important.

Over the past few decades, various cloud detection methods have been developed [11]. These methods typically rely on comparisons between clouds and background surfaces within a specified target area. Comparisons can be based on differences in single spectral bands, spectral combinations, or the temporal and spatial characteristics of clouds. Additionally, they may involve a combination of the spectral, spatial, and texture features of clouds [12].

Deep learning methods have achieved significant success in cloud detection, greatly enhancing the accuracy and generalization performance of cloud detection. Deep Convolutional Neural Networks (DCNNs) have been widely applied in cloud detection due to their ability to automatically extract high-level features from images [13]. Various metrics have been introduced in DCNNs to improve accuracy. For instance, spatial pyramid pooling mechanisms [14,15], multi-level feature fusion structures [16], attention mechanisms [17,18], physical models [19], GANs [20], and deep supervision mechanisms [21] have all been proposed for cloud detection. Furthermore, some DCNNs have been combined with remote sensing data related to geographic attributes [22], spatial features [23], and object element properties [24] to enhance cloud detection performance. Liu et al. used spectral observations from the Advanced Himawari Imager (AHI) for cloud detection based on machine learning (ML), introducing three different surface processing models to eliminate the impact of surface conditions on cloud detection [25] Shang et al. developed a cloud detection algorithm for the Cloud, Atmospheric Radiation, and Renewable Energy (CARE) dataset, which employs a threshold-based test and an additional Extremely Randomized Trees (ERT) model to detect all-day clouds in the full-disk measurements from Himawari-8 [26]. These methods have shown advanced results on challenging cloud detection or cloud-snow separation datasets.

Cloud detection is an essential step for obtaining accurate and complete ground images by removing cloud layers. However, achieving high-quality, cloud-free images is not solely dependent on effective cloud detection. Data fusion techniques play a crucial role in effectively inferring ground images under multi-cloud conditions. These techniques can integrate multiple remote sensing images to compensate for the information gaps caused by cloud contamination, thereby improving the overall quality and accuracy of the reconstructed images. Despite recent advancements in satellite imaging technology, obtaining high-quality, cloud-free images for specific times and regions remains a challenge. To address this challenge, this paper proposes a method that combines cloud detection with data fusion techniques to effectively infer ground images under cloudy conditions, ultimately improving the quality and accuracy of the reconstructed images.

### 2.2. Cloud Removal

To reduce the impact of clouds, significant efforts have been made in the field of cloud removal. The reconstruction of cloud-free images is essentially an information reconstruction process, and existing cloud removal methods can be categorized into three types based on the source of auxiliary information: spatial-based, multi-temporal, and multi-source methods [3].

Spatial-based methods utilize information from cloud-free areas within cloud images to restore pixel values in cloud shadow regions. The most basic form of these methods is interpolation [4]. For example, Meng et al. [27] used a feature dictionary learned from cloud-free areas to restore missing information patch by patch through sparse representation. When the cloud area is small, spatial-based methods can effectively remove clouds without relying on additional images, making them simple and efficient. However, as the area of cloud obstruction increases, the performance of spatial-based methods declines significantly, or they may even fail.

Multi-source methods use images obtained from one or more other sensors as auxiliary images for cloud removal. A series of studies have explored the potential of using Synthetic Aperture Radar (SAR) data as auxiliary data for cloud removal in optical images, as SAR data can penetrate cloud cover to obtain ground information beneath the clouds [28]. For instance, DSen2-CR utilized deep residual neural networks to predict the target cloud-free optical image by integrating Sentinel-1 SAR images and Sentinel-2 optical images. The Simulation Fusion GAN [9] combined SAR and damaged optical images with two generative adversarial networks to obtain cloud-free results for simulated GaoFen-2 multi-cloud data and real multi-cloud Sentinel-2 data. GLF-CR [29] integrated the contributions of Sentinel-1 SAR images in recovering reliable texture details and maintaining global consistency to reconstruct the occluded areas of Sentinel-2 optical images. With the rapid development of deep learning, the multi-source data fusion approach for cloud removal demonstrates powerful feature extraction capabilities, leading some researchers to apply CNNs to cloud removal tasks and develop SAR-based image fusion methods [30]. These methods significantly improve the utilization of information from SAR images while enhancing cloud removal effectiveness.

Multi-temporal methods utilize cloud-free images captured by the same sensor at nearby times as sources for reconstructing the cloud-obscured areas. With advances in remote sensing technology, satellites can capture images of the same location at shorter time intervals. This progress allows for the easy acquisition of multiple satellite images taken simultaneously, providing ample data support for designing effective cloud removal methods. Generally, cloud positions vary over time, and areas obscured by clouds at the same geographical location do not completely overlap at different moments. By leveraging spectral and temporal information, existing multi-temporal cloud removal methods integrate multiple mixed-cloud images to generate detailed cloud-free images. For instance, Sarukkai et al. [31] utilized a spatio-temporal generator network to approximate cloud-free Sentinel-2 images while capturing correlations among multi-temporal cloud images over the region. Huang et al. [32] proposed a Cloud Transformer generative adversarial network, which took three temporal cloud images as input and generated the corresponding cloud-free images, designed a feature extractor to maintain the weight of the cloud-free region, while reducing the weight of the multi-cloud region, and passed the extracted features to the conformer module to find the most critical representation. MCGAN [33] extended Conditional GANs (CGANs) from RGB images to multi-spectral images for cloud removal. Spa-GAN [34] introduced a spatial attention mechanism in GANs to improve information recovery in cloud regions, resulting in higher-quality cloud-free images. AE [35] adopted a convolutional autoencoder trained on multi-temporal remote sensing datasets to remove clouds. STNet [22] integrated cloud detection techniques and fused spatiotemporal features from multiple cloud images for cloud removal. CTGAN [32] introduced a transformer-based GAN for cloud removal. PMAA [36] implemented efficient cloud removal using a progressive autoencoder. Cloud removal is fundamentally an image recovery task, where complete high-quality images are reconstructed from low-quality or degraded images.

Despite these methods’ limitations, which may not maintain the spatial continuity of objects in the recovered images, multi-temporal complementary methods have proven to be effective for removing thick clouds and have been widely applied across various applications. However, further research is needed to enhance the robustness and applicability of these methods in the presence of significant spectral differences and to ensure spatial continuity in the recovered images. To address these issues, we proposed a high-performance progressive multi-scale attention autoencoder (MATT). This model effectively captures fine-grained and coarse-grained features across different scales. In the cloud removal model, we introduced a multi-scale fusion module, generating cloud mask data during the network’s operation, and combining cloud-covered images to produce cloud heat maps that enhance the ability to distinguish between land cover types. Additionally, the model reconstructed fine-grained image structures from the extracted local and global features. Experimental results indicate that this method exhibits significant advantages in both accuracy and practical effectiveness, providing an innovative solution to the cloud removal problem on the Tibetan Plateau. This enhancement facilitates better monitoring of snow cover changes and delivers accurate data support for regional water resource management.

## 3. Methodology

As shown in Figure 1, cloud removal from multi-temporal and multi-source remote sensing images mainly involves obtaining cloud-free images from multiple images. We denote the three cloudy satellite images as $\left\{X_{i}\in R^{4\times H\times W}|i=1,2,3\right\}$ here H and W represent the height and width of the images, respectively, and ‘4’ indicates four spectral channels (RGB and infrared bands). $y\in R^{4\times H\times W}$ represents the cloud-free image at the current location. We assume that for any given location $X_{i}$ changes slowly over time, and the cloud coverage within the images varies. To detect cloud-covered areas, we compute the cloud mask using multi-scale fusion, incorporating attention mechanisms to accurately differentiate between cloud-covered and snow-covered regions.

Given the input of multi-temporal satellite images $\left\{X_{i}|i=1,2,3\right\}$, we first preprocess them using a weight-sharing approach consisting of several convolutional layers with residual connections, which serves as the input to the cloud removal autoencoder. Next, the encoder in the cloud removal autoencoder utilizes a generalized representation of multi-granular resolution features, global and local multi-scale feature fusion, and efficient partitioning of multi-task information flow to extract features from both cloudy and cloud-free regions. Finally, based on the multi-scale fused features, we obtain global attention, and in the decoder, we connect the local and global features. With sufficient supplementary data, we can recover the cloud-covered areas and reconstruct the cloudy images.

### 3.1. Cloud and Snow Segmentation Module

The structure of the Cloud and Snow Segmentation Module [37] is shown in Figure 2. It consists of a feedforward stem, four feature processing modules (CF-ATT), and a projection head. Each feature extraction module contains a multi-scale feature fusion module and a feature extraction module, which extracts cloud and snow segmentation feature maps through multi-task gradient flow.

In the feature extraction module, the Multi-scale Feature Processing Module (CF-ATT) gradually extracts the deep features of the image through downsampling, while reducing the spatial resolution of the feature map, extracting multi-layer features at different scales, and improving the accuracy of cloud and snow segmentation by using the feature maps from different scales. This module not only enhances the feature representation ability of cloud and snow segmentation through the fusion of feature maps but also preserves the details of cloud and snow boundaries by learning information at various scales. Specifically, the feature extraction module first performs feature extraction on the input and then fuses feature maps from different scales to obtain richer contextual information. These feature maps are then processed through a series of convolutional layers to generate feature maps for cloud and snow segmentation for subsequent tasks.

### 3.2. High-Performance Cloud Removal Module

We have designed a novel high-performance cloud removal model to efficiently perform cloud image restoration, and the image reconstruction network uses encoders and decoders to implement the image restoration process. It takes spatiotemporal features $U_{c}\in R^{12\times H\times W}$ as input, which are obtained by concatenating $\left\{U_{i}|i,\;2,3\right\}$ along the channel dimension. Through progressive refinement, the model reconstructs cloud-covered images into cloud-free versions, enhancing image clarity and detail restoration. The cloud removal module is described in the following sections.

#### 3.2.1. Encoder

To maximize the retention of useful information, we employ a multi-scale feature extraction encoder, which captures features at different resolutions to enhance information preservation, as shown in Figure 3. By applying downsampling operations, we progressively increase the depth of the image features while reducing the spatial dimensions. This ensures a broader receptive field while retaining as much key information as possible. Specifically, we use several 3 × 3 depthwise separable convolution layers with a stride of 2 × 2, which reduces the image size and expands the receptive field. After *N* downsampling steps, *N* + 1 multi-scale features are obtained, with each feature map having different resolutions. The equation is as follows:(1)$$\left\{{F_{i}\in R}^{c\times \frac{H}{2^{i}}\times \frac{W}{2^{i}}}|i=0,\;1,2,\ldots N\right\}$$
where *C* represents the number of channels after each convolution layer, instance normalization and the ReLU activation function are applied to ensure the stability and non-linear representation of feature extraction. Finally, the extracted multi-scale features *F_i_* are fed into the multi-scale attention module for further feature fusion and processing.

#### 3.2.2. Multi-Scale Attention Module

Considering that the semantic repair of images requires deep semantic features, and many existing image inpainting network layers are often not deep enough, this paper suggests that the Multi-Scale Attention Module module is used to perform deep repair of the network layer. The module not only realizes the fusion of features obtained by convolution with different expansion rates but also realizes the fusion of shallow features and deep features, so as to ensure that the shallow feature information of the undamaged part of the image will not be lost. This paper then uses the self-attention mechanism and Selective Attention in the network. Multi-Scale Attention Module is introduced into the encoder: (1) multi-scale extraction and fusion; (2) Convolution-Self-Attention Block; and (3) Selective Attention.

Traditional encoder-decoder architectures often suffer from information loss, especially during global multi-scale feature interactions. To solve this problem, we introduce multi-scale feature extraction and fusion, as shown in Figure 4, to enhance the model’s image reconstruction ability in areas where the boundary between cloud and snow is not clear. By leveraging the complementary strengths of multi-scale features, the model effectively resolves ambiguities in cloud-snow boundaries, ensuring precise differentiation of ground objects, and thereby improving the quality and completeness of image reconstruction.

The key focus of multi-scale feature extraction lies in the hierarchical analysis of information at different scales, ensuring that the model can comprehensively capture both the detailed features and overall structure of the scene. Through a multi-branch feature extraction module, convolutional kernels with different receptive fields are dedicated to capturing global scene information and local fine-grained textures. For instance, large-scale convolutional kernels or dilated convolutions primarily focus on the overall distribution and morphological characteristics of cloud- and snow-covered areas, while small-scale convolutional kernels specialize in accurately locating boundaries, textures, and other intricate details. Additionally, by incorporating feature mappings from various receptive fields, the model achieves a comprehensive interpretation of the input imagery, providing diversity and robustness for subsequent feature processing and image reconstruction. This process aims to enhance the model’s ability to represent image features across different scales without directly performing feature fusion.

Multi-scale feature fusion is a key operation in the reconstruction of images with cloud- and snow-covered regions. Integrating features extracted from various receptive fields enables a comprehensive analysis and precise reconstruction of cloud and snow regions. First, a multi-branch structure is utilized to extract features from input images. The global branch employs dilated convolutions to capture large-scale contextual information, highlighting the overall distribution of cloud- and snow-covered areas. Meanwhile, the local branch focuses on edges and textures using standard convolution kernels, aiding in the localization of boundaries and intricate features. Furthermore, through the adaptive fusion of multi-scale features, the model can balance the importance of global and local information based on the morphological characteristics of clouds and snow, optimizing the representation of image content. During the fusion phase, a combination of feature concatenation and channel compression effectively integrates multi-branch features, enhancing the model’s semantic understanding and detail restoration capabilities for cloud and snow regions. This process not only improves the model’s perception of complex scenes with cloud and snow coverage but also significantly enhances the accuracy and completeness of image reconstruction.

In practice, the features of each branch in the F-Unit are first transformed using a standard 3 × 3 convolution, generating feature mappings at different scales. In this process, convolution operations use rotating kernels, where parameters ω and ρ represent the weight vector and bias vector of the convolution, respectively. The F-Unit shows the transformed structure of the branches.

Next, the three parallel 3 × 3 convolution kernels in the F-Unit are further fused and transformed into a single 3 × 3 convolution kernel, illustrating the fusion process of the three units in F-Unit, and showing the convolution operations in the reparametrized CF-Unit. Finally, by loading the parameters ϕ and into the 3 × 3 convolution of the CF-Unit, the lossless reparameterization process of the multi-scale feature fusion unit is completed, further enhancing the model’s ability to capture multi-scale features.

The specific calculation formula is as follows:(2)$$F_{u}=F-Unit\left(1\times 1Conv,3\times 3Conv,BN,Identitly\right)$$
(3)$$F_{u}^{\prime }=F-Unit\prime \left(3\times 3{Conv}_{1}^{\prime },3\times 3{Conv}_{2}^{\prime },3\times 3{Conv}_{3}^{\prime }\right)$$
(4)$$F_{U}{=F}_{u}^{\prime \prime }=CF-Unit\left(3\times 3{Conv}^{\prime \prime }\right)$$

Unlike CNNs, which only focus on local information, Transformer utilizes a self-attention mechanism to compute dependencies over a larger area, as shown in Figure 5, which is considered a key reason why it is superior to CNNs. However, previous research has shown that visual Transformers tend to extract local information in the shallow layers while capturing broader information in the deeper layers. This phenomenon indicates that using self-attention in the shallow layers is ineffective, as attempting to compute dependencies over larger ranges can lead to redundant calculations. Given the challenges in acquiring large-scale remote sensing datasets for cloud removal, minimizing redundant computations and parameters is essential to prevent overfitting, particularly when working with small to medium-sized datasets. To address this, we introduce self-attention in the deeper layers to effectively encode spatial information and capture long-range dependencies. To strike a balance between performance and efficiency, we implement a simplified self-attention mechanism through a convolutional modulation operation, referred to as the Convolution-Self-Attention Block (C-SA). This operation employs larger convolution kernels to avoid the time-consuming and complex issues associated with traditional attention matrix calculations, thereby enhancing the efficiency and performance of the network. Through this approach, we aim to reduce computational overhead while extracting features, enabling the model to better capture global information and contextual relationships in the cloud removal task, ultimately improving the cloud removal effect and the model’s generalization ability.
(5)$$F_{k}=F_{U}{+\alpha W}_{3}(DW{Conv}_{k\times k}(W_{1}F_{U}),⊙W_{2}F_{U})$$
(6)$$F_{g}=F_{k}+{\beta V}_{2}(V_{1}F_{k}+DW{Conv}_{k\times k}(V_{1}F_{k}))$$
where Conv represents a common 1 × 1 convolution, and DWConv represents a *k* × *k* deep convolution. In fact, *k* is set to equal to 3 and *W* is the corresponding linear transformation. *W*_1_, *W*_2_, and *W*_3_ are linear layers, α are learnable parameters, and $DW{Conv}_{k\times k}$ represents *k* × *k* deep convolution. Then, we add a residual connection after self-attention to reduce information loss. Self-attention is followed by a feedforward network (FFN), which consists of a deeply separable convolutional layer and two linear layers. *V*_1_ and *V*_2_ are linear layers, and *β* are learnable parameters. *V*_1_ and *V*_2_ are linear layers, and *β* are learnable parameters. In summary, by processing a transformer layer, we get a feature $F_{g}$ with global information.

We use global attention $F_{g}$ to perform adaptive recalibration on the input features $F_{i}$. The core idea behind this mechanism is that the information extracted by different layers of the neural network has distinct characteristics: earlier layers capture rich low-level texture features, while deeper layers extract more abstract and semantic high-level information. To effectively integrate these features, we first upsample the global attention feature $F_{g}$ using nearest neighbor interpolation so that it has the same spatial dimensions as $F_{i}$. This step ensures consistency and comparability of the information. Subsequently, we apply an affine transformation.
(7)$$F_{i}^{\prime }=\varphi \left(\sigma \left(Z_{1}\left(F_{g}\right)\right)\right)⨀Z_{2}\left(F_{i}\right)+\varphi (Z_{3}(F_{g}))$$
to obtain the feature representation $\left\{F_{i}^{\prime }\in R^{c\times \frac{H}{2^{i}}\times \frac{W}{2^{i}}}|i=0,\ldots ,N\right\}$, where $Z_{1}$,${\;Z}_{2}$,$\;Z_{3}$ are linear layers, σ denotes the Sigmoid activation function, and φ represents the nearest neighbor interpolation.

#### 3.2.3. Decoder

In our model, the global feature $O_{i}$ has a large receptive field, capable of containing rich high-level semantic information. In contrast, the local feature $F_{i+1}^{\prime }$ primarily provides low-level texture information but has a relatively small receptive field. Therefore, to effectively fuse these two types of features, we designed a local interaction module as the core component of the decoder.

The primary task of this local interaction module is to progressively restore the image resolution by utilizing the previously modulated global feature $O_{i}$ and the local feature $F_{i+1}^{\prime }$ to generate a more refined output. Specifically, we first perform convolution modulation on the global feature $O_{i}$ to enhance its ability to model local information. Then, we use the upsampled global feature as weights to obtain more robust local features.

On this basis, we concatenate the convolution-modulated $O_{i}$ with the local feature $F_{i+1}^{\prime }$, generating a refined feature representation that contains both global and local information. This process is implemented through three depth convolution layers ($D_{1}$, $D_{2}$,$\;D_{3}$), with details such as normalization layers omitted. Finally, the processed feature map will be selected as the output of the current cloud removal model, ensuring that the output image has a higher resolution and richer feature expression capability. This design not only improves the model’s expressive power but also effectively combines information from different levels, thereby enhancing the accuracy and robustness of the cloud removal process.

### 3.3. Loss Function

To optimize the cloud removal network and enhance the model’s cloud removal capabilities for generating high-fidelity cloud-free images, we calculate the *L*1 loss between the generated cloud-free images and the ground truth to align their divergences.
(8)$$\Gamma_{L1}\left(F\right)={\left|\left|y-F\left(x\right)\right|\right|}_{1}$$

Let x represent the multi-temporal cloudy images, F represent the cloud removal model, y represent the ground truth, and $F\left(x\right)$ represent the estimated cloud-free image.

## 4. Experiments

### 4.1. Data

To validate the effectiveness of the model and its components, we conducted experiments using a widely recognized dataset as well as a self-constructed dataset.

Sen2_MTC_New: This dataset [36] is created from publicly available Sentinel-2 images and contains approximately 50 non-overlapping tiles. Each tile consists of about 70 pairs of cropped 256 × 256 pixel patches, with three cloudy images corresponding to one cloud-free image. The number of channels C is 4 (RGB and infrared), and the pixel values range from 0 to 10,000.

XZ_Sen2_Dataset: To evaluate the model’s capability to remove clouds and snow in high-altitude regions, we selected data from the Nyingchi area of Tibet. Each dataset comprises three temporally close cloudy images and a cloud-free reference image, as shown in Figure 6. Due to the research objective, the selected data specifically features mixed coverage of clouds and snow in high-altitude areas. This dataset includes 16 non-overlapping remote sensing images, each cropped into 1740 patches of 256 × 256 pixels, with three cloudy images corresponding to one cloud-free image. The number of channels C is 4 (RGB and infrared). As shown in Figure 3, the dataset includes Sentinel-2 images, featuring the 10 m resolution bands of Sentinel-2, including Bands B2, B3, B4, and B8, with all band values clipped to [0, 10,000].

### 4.2. Implementation Details

To facilitate the reproducibility of experiments, this section provides a detailed overview of the hyperparameter configurations used during training, as well as the metrics employed to evaluate the experimental results.

#### 4.2.1. Training Settings

Initially, we normalize all images to the range of [−1, 1]. Multiple cloudy images are then concatenated along the channel dimension and passed through several bottleneck layers composed of convolutions for feature extraction. The downsampling and upsampling counts for both the encoder and decoder are set to 4, while the number of channels in the hidden layers is configured to 32. During the training phase, we employ the Adam optimizer, with an initial learning rate of 5 × 10^−4^ and a weight decay of 1 × 10^−5^. To manage learning rates effectively, we utilize cosine decay [23] for scheduling. The training process spans 120 epochs with a batch size of 4, and the model yielding the best Structural Similarity Index Measure (SSIM) on the validation set is selected for testing on the test set.

The experiments are conducted on an NVIDIA GeForce RTX 3090 GPU graphics card paired with an Intel Core i7 quad-core processor, running on a Windows 10 operating system. The deep learning framework used is PyTorch, and the program is implemented in Python 3.9.

#### 4.2.2. Evaluation Metrics

In all experiments, we report the Peak Signal-to-Noise Ratio (PSNR), Structural Similarity Index Measure (SSIM), Learned Perceptual Image Patch Similarity (LPIPS), and Fréchet Inception Distance (FID) for the test set, in order to assess the quality of the generated cloud-free images. It is important to note that PSNR and SSIM measure differences between images on a pixel basis, while FID and LPIPS evaluate differences based on deep feature vectors.

Peak Signal-to-Noise Ratio (PSNR) is a traditional Image Quality Assessment (IQA) metric, where higher PSNR values typically indicate higher image quality. Equation (9) can be defined as follows:(9)$$\mathrm{PSNR}=10\text{×}\log_{10}\left(\frac{MAX^{2}}{MSE}\right)$$
(10)$$MSE=\frac{\sum_{i=1}^{N}{\left(y_{i}-\overset{\wedge }{y_{i}}\right)}^{2}}{N}$$
where $y_{i}$ and ${\hat{y}}_{i}$ are the actual and simulated values for the *i*-th pixel, and *N* is the number of pixels. Where *MAX* represents the maximum pixel value of the image, in this context, *MAX* is 255. *MSE* is an acronym for Mean Squared Error and can be defined by Equation (10).

The Structural Similarity Index (SSIM) measures the structural similarity between a real image and a simulated image and is defined by Equation (11):(11)$$SSIM\left(x,y\right)=\frac{\left(2\mu_{x}u_{y}+c_{1})(2\sigma_{xy}+c_{2}\right)}{\left(\mu_{x}^{2}+\mu_{y}^{2}+c_{1})(\sigma_{x}^{2}+\sigma_{y}^{2}+c_{2}\right)}$$
where $\mu_{x}$ is the mean of x, $\mu_{y}$ is the mean of y, $\partial_{x}$ is the variance of x, $\partial_{y}$ is the variance of y, $\partial_{xy}$ is the covariance of x and y, $c_{1}=(k_{1}L{)}^{2}$ and $c_{2}=(k_{2}L)$ are constants to maintain stability, and L is the dynamic range of pixel values. The SSIM value ranges from −1 to 1, where a value of 1 indicates identical images.

Learned Perceptual Image Patch Similarity (LPIPS) is a metric used to measure the perceptual similarity between two images. Unlike traditional metrics such as PSNR and SSIM, LPIPS is learned through deep learning methods and can better simulate human visual perception. A smaller LPIPS value indicates greater similarity between the two images. Given a reference image patch x and a noisy distorted image patch $x_{0}$, the perceptual similarity metric is formulated as follows:(12)$$d(x,x_{0})=\sum_{l}\frac{1}{H_{l}W_{l}}\sum_{h,w}||w_{l}⊗({\overset{\Lambda }{y}}_{hw}^{l}-{\overset{\Lambda }{y}}_{0hw}^{l})|{|}_{2}^{2}$$
where d is the distance between $x_{0}$ and x. Extract the feature stack from the L layers and perform unit normalization along the channel dimension. Scale the activation channels using the vector $w_{l}$ and finally, calculate the $L_{2}$ distance. Then, average over the spatial dimensions and sum over the channels.

The Fréchet Inception Distance (FID) is a metric used to evaluate the performance of generative models, particularly in the context of Generative Adversarial Networks (GANs). It aims to measure the difference between the distributions of generated images and real images, indicating the quality and diversity of the generated images. A lower *FID* value signifies that the generated images are closer to the distribution of real images, and thus, it is generally considered indicative of a better generative model. The Equation is calculated as follows:(13)$$FID=||\mu_{r}-\mu_{g}|{|}^{2}+T_{r}(\sum_{r}+\sum_{g}-2({\sum_{r}\sum_{g})}^{1/2})$$

$\mu_{r}$: Calculation of the Mean of Real Image Features

$\mu_{g}$: Generating Image Feature Data

$\Sigma_{r}$: Covariance Matrix of Real Images

$\Sigma_{g}$: Covariance Matrix of Real Images

Compared to traditional image quality assessment metrics (PSNR and SSIM), FID and LPIPS align more closely with human visual perception. Moreover, LPIPS can almost be considered a localized version of the FID metric, as it tends to exhibit minimal variation even when FID scores fluctuate significantly. Therefore, when LPIPS scores approach similar values, we prioritize the FID scores for assessment.

### 4.3. Cloud Removal Results

In this section, we conduct experiments using images from the Sen2_MTC_New and XZ_Sen2_Dataset datasets to evaluate the effectiveness of the proposed cloud image reconstruction method. In most cases, as long as the local weather is not excessively cloudy, the proposed method can effectively reconstruct the cloud-free areas of the original images. We tested the multi-cloud image algorithm on land images from different regions. We used three different terrains: grasslands, mountains, and fields, to assess the performance of the cloud image algorithm across varying landscapes. All terrains were obtained from Sentinel-2, with images composed of bands 4, 3, and 2 to display natural true color images. Despite significant local variations in spectrum and radiation, along with the complexity of the regions and notable elevation changes, the proposed reconstruction method demonstrated its effectiveness, successfully recovering most of the information obscured by clouds.

As shown in Figure 7, the impact of removing thick clouds on three temporal images in the agricultural scene is illustrated. The figure demonstrates that our method effectively restores non-overlapping areas from different time points. However, as indicated by the red box, the texture recovery in overlapping cloud-covered regions varies. Overall, the restoration results of this method are more realistic and resemble the original images, with a better restoration effect on specific details.

As shown in Figure 8, this model demonstrates the effectiveness of thick cloud removal in green land scenes, comparing the reconstructed images with the original images. The green land undergoes minimal change across the three time points, resulting in slight differences in the recovery outcomes. It is evident that the restoration of water areas is less effective, with average detail recovery capabilities. The edges of rivers appear blurred, leading to unnatural visual textures and the loss of small details.

Through Figure 9 and Figure 10, we can observe that removing clouds in mountainous areas with complex textures and snow cover presents significant challenges. Figure 9 illustrates the cloud removal reconstruction effect in snow-covered mountainous regions. The intricate terrain contains a wealth of details, making it difficult to fully restore all features. The results indicate that the repair effect is better in areas with light cloud cover, where the mountainous texture information is well preserved. In Figure 10, depicting areas with heavy cloud cover, as shown in the yellow box, there are issues with incomplete recovery of the cloud-free images at different time points, leading to some loss of features and color distortion in certain regions. However, the extent of the snow-covered areas has been well preserved.

This experiment evaluated our model’s performance in cloud removal for high-altitude snow-covered areas. By adjusting various modules within the model, we obtained cloud mask images for cloud removal in snow-covered regions. The integration of attention modules allowed for effective localization of most cloudy areas during the removal process, while retaining regions with snow that are similar to clouds, facilitating subsequent studies on changes in snow-covered areas.

## 5. Discussion

### 5.1. Contribution of This Study

We designed a series of ablation experiments to verify the effectiveness of the MATT component. We conducted the ablation experiments on the XZ_Sen2_Dataset. Table 1 presents the results of the quantitative experiments, where we can observe that enhancing the attention on cloudy areas significantly improves the accuracy of cloud removal in snow-covered regions of the plateau. Additionally, we explored the impact of the multi-scale fusion module on cloud removal results. From the comparison of results in Table 1, we found that the multi-branch fusion module yielded higher precision. This indicates that each of our modules is effective, demonstrating that our proposed method has strong performance.

Since the proposed method is based on temporal recovery, it assumes that there are similar or stable spectral characteristics at different time points in the target area. While our method shows typical performance, it still exhibits certain limitations. Firstly, despite our efforts to minimize the impact of temporal differences on the reconstructed images, significant changes in snow-covered areas remain evident. Secondly, the reconstruction of texture information in complex mountainous environments in plateau regions has not been preserved with high precision. Lastly, although our method is specifically designed for cloud removal tasks, it has not undergone comprehensive validation across multiple tasks.

### 5.2. Ablation Experiments

#### 5.2.1. Multi-Scale Fusion Module

We investigated the impact of various multi-scale feature fusion methods on cloud removal performance, as shown in Table 1. Compared to the summation operation, multi-branch feature fusion emerged as the optimal feature fusion strategy, ultimately enhancing PSNR by 0.219 and SSIM by 0.039. It is noteworthy that while multi-branch feature fusion improves the accuracy of the final reconstruction, it also increases computational complexity.

#### 5.2.2. Self-Attention

As presented in Table 1, we examined the effects of various self-attention strategies on cloud removal performance. We compared C-MSA and W-SA (Window-Self-Attention) and concluded that the model with the added convolutional self-attention module achieved the best cloud removal results.

#### 5.2.3. Selective Attention (SA)

In our study of the impact of selective attention on cloud removal performance, we emphasized the differences between clouds and snow by overlaying the cloud mask as one of the input layers on the cloudy images. The results indicated that employing the selective attention module can enhance cloud removal performance. When the cloud mask module is added as an input layer to strengthen selective attention, it improves the segmentation of clouds and snow in the attention map. The shallow feature extraction in the network contains some redundant information, and the selective attention mechanism effectively filters out this useless information while enhancing useful information, resulting in more refined feature representations. By incorporating the cloud mask layer in the selective attention module, the generated cloud attention map for cloud and snow segmentation becomes more accurate. Accuracy evaluation metrics indicate that the proposed method achieves a cloud detection accuracy greater than 0.9, with low error and miss detection rates. This level of accuracy is sufficient for reconstructing cloud-free images. It is important to note that detection accuracy may be influenced by cloud thickness; thicker clouds generally result in higher accuracy.

SA enables cloud region localization. To verify the distinction capability of SA for cloud and snow, it is evident that the added cloud mask module significantly improves the segmentation of cloud and snow. As shown in Figure 11, the combination of attention modules generates an attention map that successfully fits the regional distribution of clouds, which proves the rationality of the cloud removal model. 

### 5.3. Comparison with State-of-the-Art Techniques

We provide a comparison of the cloud removal performance and efficiency of our proposed model against existing models such as CTGAN [32], Pix2Pix [38], DENen2-CR [27], UnCRtainTS [39], PMAA [36], AttentionGAN [40], and HS2P [41] on the Sen2_MTC_New and XZ_Sen2_Dataset datasets. As shown in Table 2, our model consistently achieves superior PSNR and SSIM scores, demonstrating the effectiveness of our approach. On both benchmark datasets, the performance of our model significantly outperforms previous models and methods.

As shown in Figure 12, we have selected four sets of images to illustrate the effects of different cloud removal methods and to show how our approach compares to existing methods. We observed that our model had the highest reconstruction accuracy in high-altitude cloud and snow cover areas compared to other models.

MATT achieved consistent results across all evaluation metrics for cloud removal, demonstrating the full potential of our proposed method; specifically, MATT’s PSNR = 29.095, SSIM = 0.897, FID = 125.328, and LPIPS = 0.356. In contrast, the performance values of existing models were significantly lower. According to Table 2, our method performed well, particularly as it surpassed methods based solely on SAR (CGAN) when using SAR-optical fusion. The possible reasons are as follows: SAR images primarily capture structural characteristics of surfaces. Therefore, the CGAN relying solely on SAR images can restore major geometric information but struggles to recover local color variations. The PMAA method, which utilizes multi-temporal variations and attention mechanisms for cloud differentiation, can remove clouds, but using only cloud images in remote sensing yields visually plausible yet misleading information, as thick cloud layers completely obscure the surface and result in lost information. In contrast, SAR-optical fusion methods can leverage SAR to provide occlusion information while using cloud images for rich color and texture, leading to better outcomes.

From the visualization results, we observed that two-step fusion methods (i.e., Sim-Fus-GAN, DRIBs-GAN, and ours) restored more details than one-step fusion methods (i.e., SAR-Opt-cGAN, DSen2-CR, TF-CRNet, and GLF-CR). We explain this by stating that a single fusion network simultaneously learns cloud segmentation, SAR-to-optical transformation, and global consistency maintenance, which have differing task characteristics, making it challenging to ensure optimal performance across all tasks. For example, if the conversion from SAR to optical is not well learned, the recovered texture and details will be compromised. In contrast, two-step algorithms learn the SAR-to-optical conversion separately, making full use of SAR information. However, they still face challenges such as the heterogeneity between SAR and optical features and the imbalance in handling cloud and non-cloud regions. Our method alleviates these issues and achieves higher performance. Overall, the multi-scale progressive cloud removal model for high-altitude snow-covered mountainous areas outperforms all existing multi-temporal cloud removal models. Compared to existing SAR-integrated cloud removal models, MATT achieved significant performance improvements on both datasets. These results indicate that the MATT model excels in both accuracy and efficiency for high-altitude mountain regions with snow and cloud cover.

### 5.4. Limitations and Future Works

While the proposed method for cloud removal in snow-covered areas of complex plateau terrain still has some limitations, it demonstrates good effectiveness in removing cloudy regions while preserving snowy areas. However, there are still several restrictions persist. Firstly, due to the structural limitations of the model, MATT does not fully consider the attention mechanism across different channels, as there are variations in the cloud penetration capabilities of different spectral bands in optical images. This also contributes to the model’s better performance in visual feature recovery. Secondly, while the model learns global features, the concept of global features pertains to the 256 × 256 slices input into the model, which may not accurately describe the characteristics of panoramic remote sensing images.

In future work, we aim to extensively create training datasets tailored to different satellite data sources and investigate data from various origins and resolutions, further validating the generalization capability of the MATT model. Additionally, efforts should be made to enhance the capability to differentiate clouds from snow, as well as to improve the accuracy of restoration, thereby minimizing the temporal factors’ impact on image reconstruction.

## 6. Conclusions

This paper addresses the substantial cloud removal errors that can occur in scenarios where snow and clouds coexist, proposing a multi-attention progressive cloud removal model tailored to snow-covered mountainous areas in the Tibetan Plateau. Compared to other baselines, the integration of multi-scale fusion, global attention, and selective attention modules significantly reduces errors in easily confusable areas. The results demonstrate that this network achieves a cloud removal accuracy of PSNR = 29.095, SSIM = 0.897, FID = 125.328, and LPIPS = 0.356 for high-altitude snow-covered mountainous regions. The experiments indicate that this model is significant for establishing cloud removal systems in practical deployments and shows promise for superior performance in large-scale and batch-processing tasks. Moreover, due to its effectiveness and lightweight nature, we believe the introduced modules can be seamlessly adapted to other related problems, such as image restoration (e.g., for rain, snow, fog, and noise removal) and other generative tasks. In the course of the study, we only studied the Nyingchi region of Tibet, which lacks a certain representativeness and does not indicate its full applicability in other regions. At the same time, for the de-cloud algorithm, multi-source data, such as SAR data, can be added in the follow-up research to achieve a better de-cloud reconstruction effect.

## Figures and Tables

**Figure 1 sensors-24-07848-f001:**
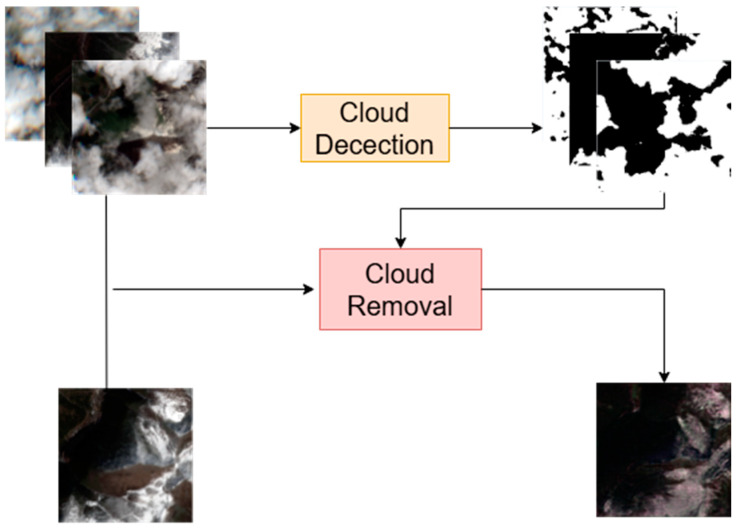
A brief explanation of the input and output for cloud detection and removal data is as follows: three cloudy remote sensing images from different periods, their corresponding cloud-snow segmentation masks, and a cloud-free reference image are processed through the cloud removal model to generate reconstructed cloud-free images.

**Figure 2 sensors-24-07848-f002:**
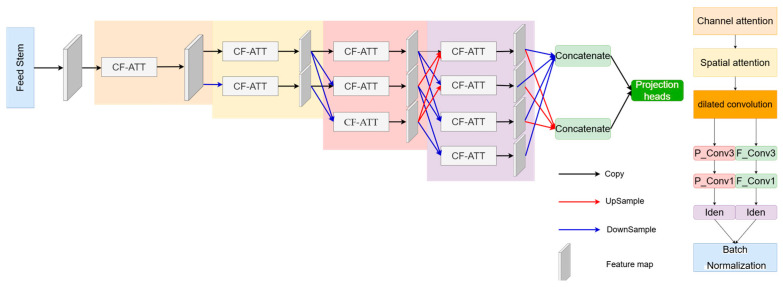
A module that uses multiple feature processing modules to segment clouds and snow.

**Figure 3 sensors-24-07848-f003:**
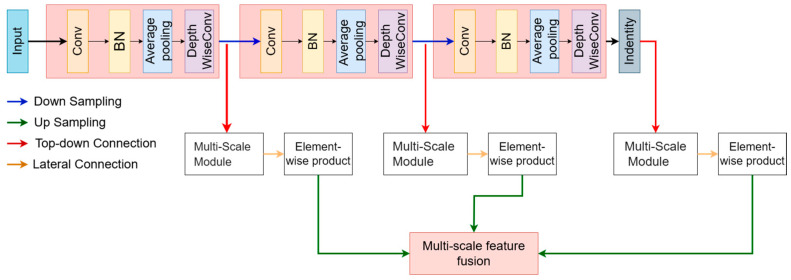
In the encoder, we downsample the input image N times. Then, multi-scale features are fused through average pooling and multi-branch convolutions. Multi-scale feature fusion layer processes the fused features to obtain global attention for modulating the multi-scale features. During the reconstruction process, we use a local interaction module to recover more details.

**Figure 4 sensors-24-07848-f004:**
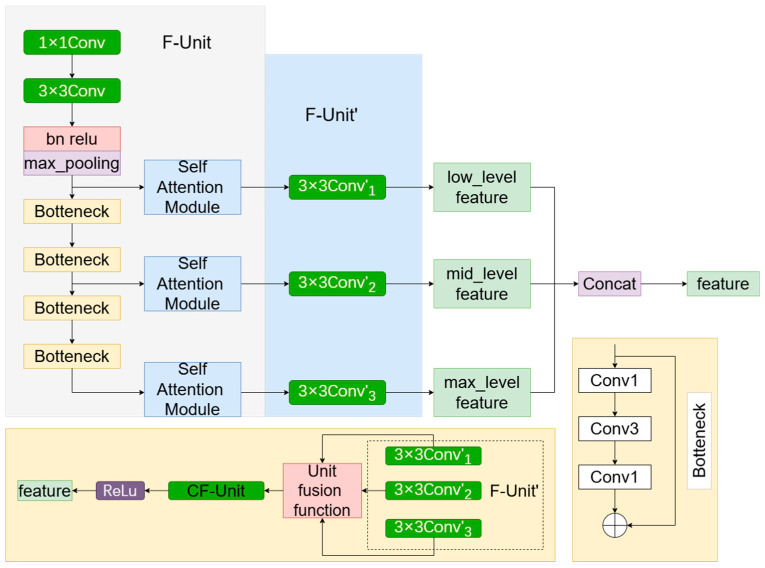
Multiscale Attention Module CF-ATT. It includes multi-scale feature extraction and multi-scale feature fusion modules, as well as reparameterization.

**Figure 5 sensors-24-07848-f005:**
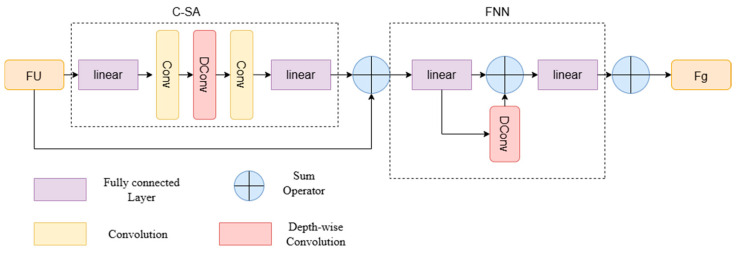
Convolution-Self-Attention Block and feedforward network.

**Figure 6 sensors-24-07848-f006:**
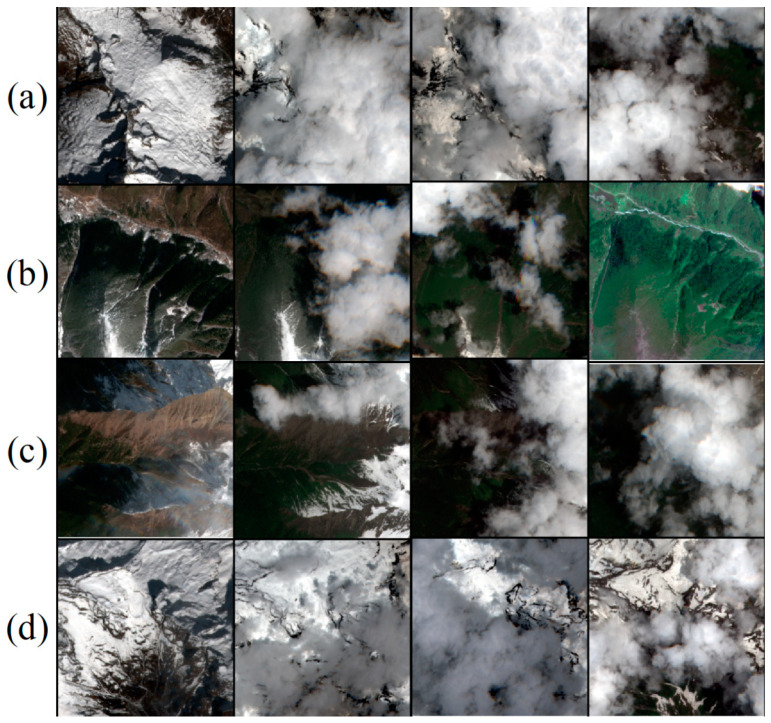
The XZ_Sen2_Dataset contains representative instances of cloud-snow mixed data from high-altitude areas. Each image data set includes three cloud-covered images taken at different times and a corresponding cloud-free reference image. (**a**–**d**) represent images from different seasons and cloud amounts.

**Figure 7 sensors-24-07848-f007:**
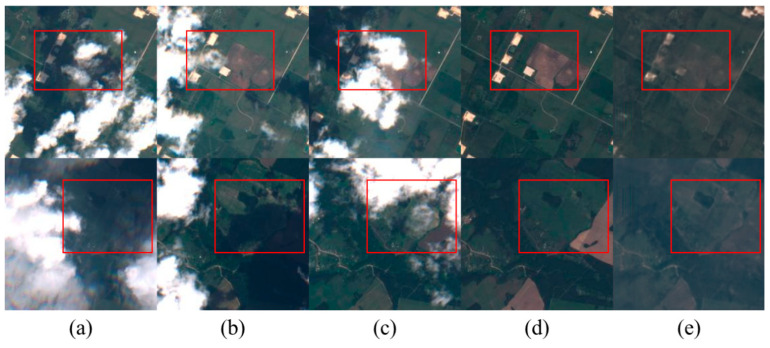
Cloud removal experimental results in agricultural scenes. (**a**–**c**) are cloud-free images from different time periods, (**d**) represents cloud-free reference images, and (**e**) represents decloud-free reconstructed images. The red box indicates the key decloud-de-rebuilding area.

**Figure 8 sensors-24-07848-f008:**
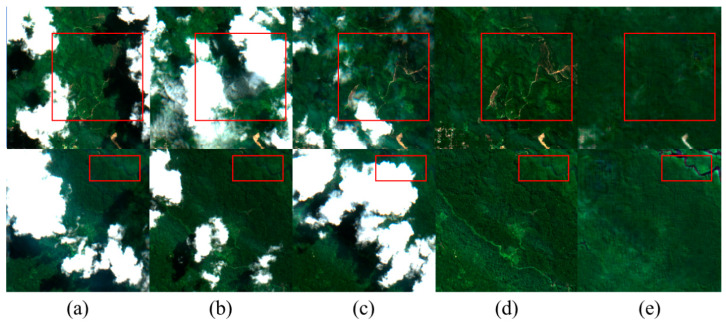
Cloud removal experimental results in the green land scene. (**a**–**c**) are cloud-free images from different time periods, (**d**) represents cloud-free reference images, and (**e**) represents decloud-free reconstructed images. The red box indicates the key decloud-de-rebuilding area.

**Figure 9 sensors-24-07848-f009:**
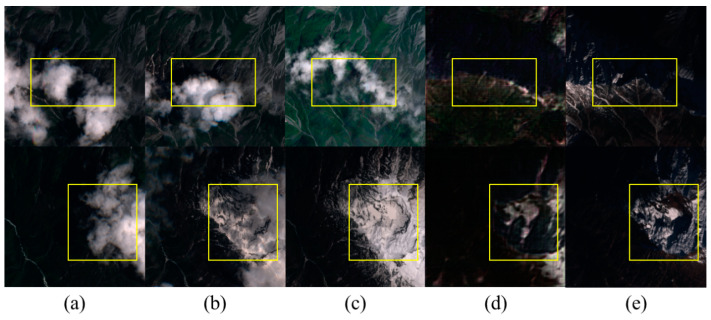
Cloud removal results in mountainous areas with light cloud cover and snow. (**a**–**c**) are cloud-free images from different time periods, (**d**) represents cloud-free reference images, and (**e**) represents decloud-free reconstructed images. The yellow box indicates the key decloud-de-rebuilding area.

**Figure 10 sensors-24-07848-f010:**
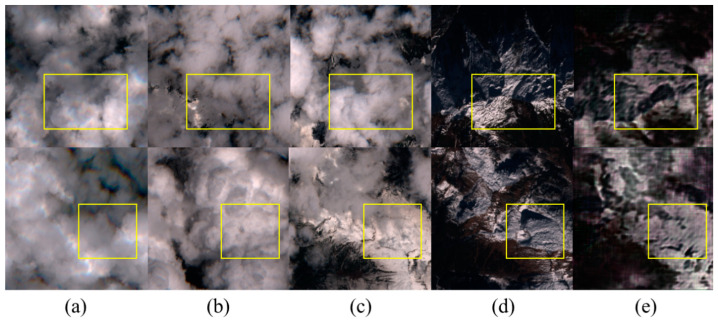
Cloud removal results in mountainous areas with heavy cloud cover and snow. (**a**–**c**) are cloud-free images from different time periods, (**d**) represents cloud-free reference images, and (**e**) represents decloud-free reconstructed images. The yellow box indicates the key decloud-de-rebuilding area.

**Figure 11 sensors-24-07848-f011:**
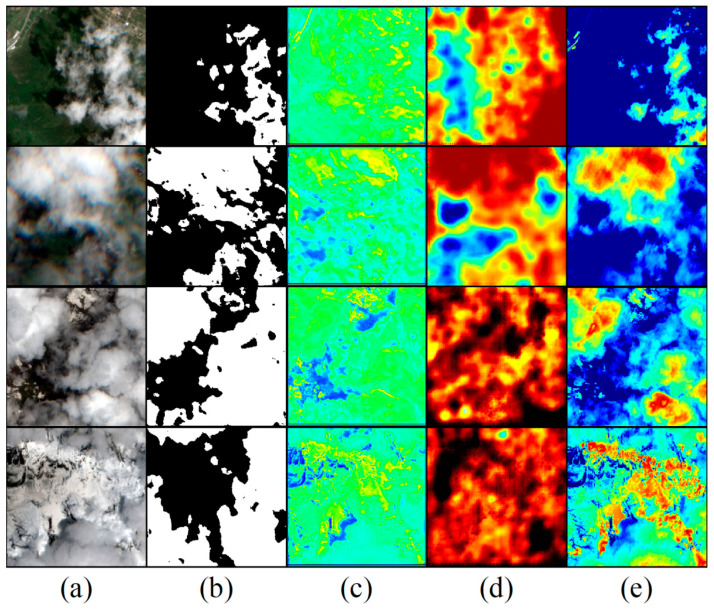
Attention maps of different attention models: (**a**) cloud coverage map; (**b**) mask for cloud and snow segmentation in snow-covered mountainous areas; (**c**) C-MSA attention map; (**d**) attention map with added selective attention; (**e**) LIM map.

**Figure 12 sensors-24-07848-f012:**
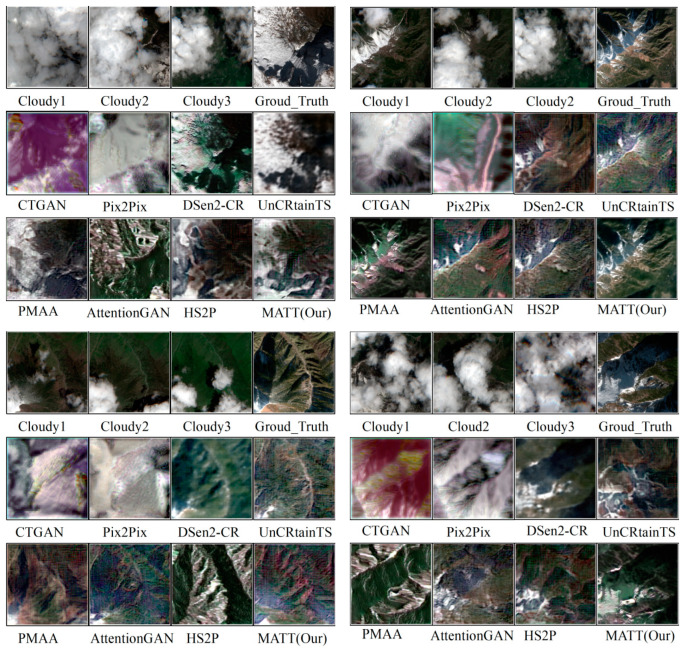
In the case of four different levels of cloud coverage, each data set represents cloud-covered images and cloud-free images, along with the image reconstruction results of cloud-snow covered areas using different cloud removal methods.

**Table 1 sensors-24-07848-t001:** Ablation study on the XZ_Sen2_Dataset. A “√” indicates that the component was used, while a “×” indicates that it was not.

Multi-Scale Fusion		Attation-Self		SA	PSNR	SSIM
Sum	Multi-Branched	W-SA	C-SA			
×	×	×	×	×	28.189	0.786
√	×	×	×	×	28.455	0.795
×	√	×	×	×	28.674	0.834
×	√	√	×	×	28.727	0.851
×	√	×	√	×	28.907	0.889
×	√	×	√	√	29.095	0.897

**Table 2 sensors-24-07848-t002:** Quantitative comparison of cloud removal performance of MATT with existing models on Sen2_MTC_New and XZ_Sen2_Dataset.

Method	Sen2_MTC_New				XZ_Sen2_Dataset			
	PSNR	SSIM	FID	LPIPS	PSNR	SSIM	FID	LPIPS
CTGAN	16.223	0.449	161.334	0.501	21.372	0.448	193.470	0.504
Pix2Pix	16.872	0.566	143.762	0.423	22.980	0.488	169.374	0.467
DSen2-CR	18.995	0.607	134.672	0.513	26.796	0.833	133.387	0.330
UnCRtainTS	19.702	0.628	118.737	0.364	26.419	0.824	138.634	0.402
PMAA	18.331	0.615	125.456	0.372	27.289	0.837	130.856	0.367
AttentionGAN	18.760	0.600	122.372	0.398	27.343	0.877	134.538	0.333
HS^2^P	19.642	0.686	99.876	0.319	28.052	0.862	130.279	0.336
MATT (Our)	19.767	0.688	98.407	0.315	29.095	0.897	125.328	0.356

## Data Availability

The data that support the findings of this article are not publicly available due to privacy, and ethical concerns.

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
