# Peer review of "Cloud Removal in the Tibetan Plateau Region Based on Self-Attention and Local-Attention Models"

_sensors, 2024, doi:10.3390/s24237848_

Round 1

Reviewer 1 Report

Comments and Suggestions for Authors

The manuscript by Zhao et al. introduced a cloud removal model for optical remote sensing images using a combination of self-attention and local attention model. The model is applied for images of the Tibetan Plateau region, where the surface condition is complex due to snow and ice coverage. The model achieves reasonable accuracy, and is important and useful for downstream applications. The manuscript is well organized and presented, and could be considered for publication after a minor revision.

1.  Section 2.1 introduced some of the cloud detection models based on DCNNs, while missed much more. There are a large number of interesting and powerful models of similar kinds. Some of the examples include https://doi.org/10.1016/j.rse.2023.113957 and  https://doi.org/10.1007/ s00376-021-0366-x, and those unique models should be considered or mentioned in the manuscript.  

2. Figures 2 and 3 show quite general structures of networks for cloud and snow segmentation and cloud removal model. Such general frameworks are less useful for readers interested in this study. The figures are suggested to be improved by considering the particular settings of this work, and make a particular plot for the method in this paper. This also applies to Figures 4 and 5.

3. The discussions in Section 3 are also relatively general. The unique treatment of this study is suggested to be better presented. 

4. The discussions on the results in Section 4.3 are qualitative, and can the accuracy of the examples be quantified? Meanwhile, how is the current model comparable to similar models, and such comparison may be briefly presented in the paper.

5. There are some English typo or grammar errors in the manuscript, and careful proofread is suggested as well. 

Author Response

Point 1: Section 2.1 introduced some of the cloud detection models based on DCNNs, while missed much more. There are a large number of interesting and powerful models of similar kinds. Some of the examples

include https://doi.org/10.1016/j.rse.2023.113957 and  https://doi.org/10.1007/ s00376-021-0366-x, and those unique models should be considered or mentioned in the manuscript.  

Response 1: Thank you so much for pointing this out. We studied both papers and cited their models as references. Thank you so much again for this valuable comment.

Point 2: Figures 2 and 3 show quite general structures of networks for cloud and snow segmentation and cloud removal model. Such general frameworks are less useful for readers interested in this study. The figures are suggested to be improved by considering the particular settings of this work, and make a particular plot for the method in this paper. This also applies to Figures 4 and 5.

Response 2: Thank you so much for pointing this out. Thank you for the modification, we have modified Figure 2, Figure 3, Figure 4, Figure 5 to supplement the settings that are more suitable for the work. Thank you so much again for this valuable comment.

Point 3: The discussions in Section 3 are also relatively general. The unique treatment of this study is suggested to be better presented. 

 Response 3: Thank you so much for pointing this out. We have made simple modifications to the third section in order to better present the unique treatment of this study. Thank you so much again for this valuable comment.

Point 4: The discussions on the results in Section 4.3 are qualitative, and can the accuracy of the examples be quantified? Meanwhile, how is the current model comparable to similar models, and such comparison may be briefly presented in the paper.

Response4: Thank you so much for pointing this out. I'm sorry that my expression may have misled you, but we've removed the quantitative adjectives, and for comparisons with other models, we've calculated the accuracy metric. Thank you so much again for this valuable comment.

Point 5: There are some English typo or grammar errors in the manuscript, and careful proofread is suggested as well. 

Response5: Thank you so much for pointing this out. I did a detailed examination of the paper, correcting the grammatical errors in it. Thank you so much again for this valuable comment.

Reviewer 2 Report

Comments and Suggestions for Authors

Dear authors, congratulation for your work. I liked many parts of your article but I would need to ask you to look at my comments for improving the manuscript.

Author Response

Point 1: Line 16 – The word “Considering” is by mistake with capital “C”?

Response 1: Thank you so much for pointing this out. We have made a fix to this syntax error.

Thank you so much again for this valuable comment. 

Point 2: Line 74-76 – Did the authors calculate the average annual cloud coverage of 69.5% on the Tibetan Plateau by themselves? If yes, please indicate how you calculated it. If not, please give the reference where this percentage number is mentioned. 

Response 2: Thank you so much for pointing this out. The original description is not accurate, and the average annual maximum cloud cover on the Tibetan Plateau is 87%.References are ZHANG Jingshu, JING Linhai, WANG Siyuan, 2023. Spatial and Temporal Variations of Cloud Parameters over the Qinghai-Xizang Plateau during the Past Two Decadesï¼»Jï¼½.Plateau Meteorology, 42 (5): 1107-1118. DOI: 10. 7522/j. issn. 1000-0534. 2022. 00081 Thank you so much again for this valuable comment. 

Point 3: Line 80 – What are the different scales of fine-grained and coarse-grained features mentioned here? Are those spatial scales? If possible, give some examples indicating numbers. 

Response 3: Thank you so much for pointing this out. In the downsampling part of the model, the multi-scale feature fusion block (MFF_Block) is used to gradually reduce the resolution of the input image and extract multi-layer features. The multi-scale feature fusion block is used to gradually restore the image resolution, and at the same time, the detailed feature recovery is realized by combining the features of the jump connection. The downsampling layer downsamples the feature maps of different resolutions during the upsampling process for multi-resolution supervision and final output fusion. Both fine and coarse particles refer to the feature maps of different resolutions in the downsampled layer. Thank you so much again for this valuable comment.

Point 4: Line 81 -  Explain/Define the “F-Unit” terminology. 

Response 4: Thank you so much for pointing this out. The definition of a multi-scale feature fusion unit (F-Unit) in this model is defined in line 267. Thank you so much again for this valuable comment.

Point 5: Line 78-89 -  This part should need to be moved to another section. Please move it to an existing other section or make a new section which will be more suitable than the “Introduction”. 

Response 5: Thank you so much for pointing this out. We shifted lines 78-89 to the second

part to make the article structure smoother. Thank you so much again for this valuable comment.

Point 6: The name of Section 2 “Related Works” seems not optimal to me. I would recommend the authors to merge Section 2 with the “Introduction” section, as it all refers to literature studies. 

Response 6: Thank you so much for pointing this out. I'm going to think about it and adjust accordingly. Thank you so much again for this valuable comment.

Point 7: Line 113 – What is GANs? 

Response 7: Thank you so much for pointing this out. The method of deep learning used in Ref. 21 consists of two neural networks: a generator and a discriminator. The generator is responsible for generating fake data, while the discriminator is responsible for distinguishing real data from fake data. These two networks compete with each other, constantly optimizing and ultimately producing results that are getting closer and closer to real data. Thank you so much again for this valuable comment.

Point 8: Figure 1 seems to me confusing. After the cloud removal, it seems to me that cloud contamination is still there as it is illustrated in the bottom right picture of Figure 1. 

Response 8: Thank you so much for pointing this out. It is possible that the chosen picture does not have obvious effects that make you wonder, I have modified it, thank you again for your comments on the revisions.

Point 9: Line 173 - What do you mean “rely on (GANs)“ ? 

Response 9: Thank you so much for pointing this out. It's because my description wasn't accurate enough, so I'll change it to "In addition, the current existing multi-phase cloud removal methods use generative adversarial networks (GANs) to improve the effectiveness of image reconstruction by introducing adversarial losses into the training process." Thank you so much again for this valuable comment.

Point 10: Section 3.1 would require some more detailed introduction. Please elaborate more on the explanation of the modules before showing Figure 2. 

Response 10: Thank you so much for pointing this out. I've made changes to Figure 2 and explained the modules in it. Thank you so much again for this valuable comment.

Point 11: Figure 2 lacks of the explanation in the caption. 

Response 11: Thank you so much for pointing this out. We've renamed Figure 2 to make it more consistent with the model. Thank you so much again for this valuable comment.

Point 12: Line 297-299 – In this sentence the authors refer to previous research. Please provide a reference for those findings. 

Response 12: Thank you so much for pointing this out. Thank you so much again for this valuable comment.

Point 13: Line 320 – Do you mean “we add”? 

Response 13: Thank you so much for pointing this out. The meaning of this phrase is to add a residual connection after the self-attention module to reduce information loss. We changed the original text to read: Then, we add a residual connection after self-attention to reduce information loss. Thank you so much again for this valuable comment.

Point 14: Line 337 and 417 – There are some problems with the spacings in those lines. 

Response 14: Thank you so much for pointing this out. We've made changes where there is spacing. Thank you so much again for this valuable comment.

Point 15: Line 426 – Something looks not right with the punctuation. 

Response 15: Thank you so much for pointing this out. We had some problems with the formatting while using the formula editor, and we made changes to the problematic areas. Thank you so much again for this valuable comment.

Point 16: Line 432 – The symbol x0 seems like it is superscript. This is an issue of several mathematical symbols in the manuscript. Please revise all mathematical symbols in the text so as they appear normal. 

Response 16: Thank you so much for pointing this out. We've made changes to the problematic areas. Thank you so much again for this valuable comment.

Point 17: Line 435 – “where” with lower case. 

Response 17: Thank you so much for pointing this out. Let's change "where" to lowercase. Thank you so much again for this valuable comment.

Point 18: Figures with subfigures (like Figure 6 with subfigures 6a, 6b, 6c, 6d) need some further explanation in the text but also at the caption. 

Response 18: Thank you so much for pointing this out. A simple explanation has been added to the text, and each set of images includes three cloud images of similar time and one reference image of cloud absence for experimental purposes. Thank you so much again for this valuable comment.

Point 19: Table 2 - How did you calculate the numbers mentioned in Table 2? Please include this important information in the text. 

Response 19: Thank you so much for pointing this out. The information in the table is the accuracy evaluation index data calculated by the batch calculation code. Thank you so much again for this valuable comment.

Point 20: Figure 12 -  What exactly does this Figure show? It has too compact information hidden in all those images. Please include explanation into the text. 

Response 20: Thank you so much for pointing this out. We've resized the image so that it's less compact, and we've added a description to the body. Thank you so much again for this valuable comment.

Point 21: Line 625-630 -  Can you do already for this article what you propose as future work in this paragraph? What are the limitations or blockages that you could not finalize the relevant work so far? 

Response 21: Thank you so much for pointing this out. We've added plans for future work: In the course of the study, we only studied the Nyingchi region of Tibet, which lacks a certain representativeness and does not indicate the full applicability in other regions. At the same time, for the de-cloud algorithm, multi-source data, such as SAR data, can be added in the follow-up research to achieve a better de-cloud reconstruction effect. Thank you so much again for this valuable comment.

Point 22: Line 639 – What do you mean by “extensive experiments”? Like measurement field experimental campaigns? 

Response 22: Thank you so much for pointing this out. Due to a misunderstanding caused by a translation error, it should be "Experiments indicate that this model is significant for establishing cloud removal sys-tems in practical deployments and shows promise for superior performance in." large-scale and batch-processing tasks.” Thank you so much again for this valuable comment.

Point 23: Data availability -  Sentinel-2 data should be publicly available through the Copernicus data space ecosystem website Copernicus Data Space Ecosystem | Europe's eyes on Earth. What are the ethical reasons that you cannot make public your datasets? 

Response 23: Thank you so much for pointing this out. As this is a dataset for parts of the Tibetan Plateau, the complete dataset is being developed and is expected to be made public when the full results are released in the future. Thank you so much again for this valuable comment.
